# Association of remoteness and ethnicity with major amputation following minor amputation to treat diabetes-related foot disease

Chanika Alahakoon[1,2], Shivshankar Thanigaimani[1,3], Tejas P. Singh[1,4,5], Aaron Drovandi[1,6], James Charles[7], Malindu Fernando[1,5,8], Peter A. Lazzarini[9,10], Joseph V. Moxon[1,3], Jonathan Golledge [1,3,4]*

1 Queensland Research Centre for Peripheral Vascular Disease, College of Medicine and Dentistry, James Cook University, Townsville, Queensland, Australia, 2 Faculty of Medicine, University of Peradeniya, Peradeniya, Sri Lanka, 3 The Australian Institute of Tropical Health and Medicine, James Cook University, Townsville, Queensland, Australia, 4 The Department of Vascular and Endovascular Surgery, Townsville University Hospital, Townsville, Queensland, Australia, 5 The Department of Vascular and Endovascular Surgery, John Hunter Hospital, Newcastle, New South Wales, Australia, 6 School of Biomedical Sciences, Faculty of Biological Sciences, University of Leeds, Leeds, West Yorkshire, United Kingdom, 7 First Peoples Health Unit, Griffith University, Brisbane, Queensland, Australia, 8 Faculty of Health and Medicine, School of Health Sciences, University of Newcastle, Newcastle, New South Wales, Australia, 9 School of Public Health and Social Work, Queensland University of Technology, Brisbane, Queensland, Australia, 10 Allied Health Research Collaborative, Metro North Hospital and Health Service, Brisbane, Queensland, Australia

* jonathan.golledge@jcu.edu.au

**Data Availability Statement:** The underlying data is restricted for ethical and legal reasons. To obtain Queensland Health data, researchers need to seek

## Abstract

### Introduction

Minor amputation is commonly needed to treat diabetes-related foot disease (DFD). Remoteness of residence is known to limit access to healthcare and has previously been associated with poor outcomes. The primary aim of this study was to examine the associations between ethnicity and remoteness of residency with the risk of major amputation and death following initial treatment of DFD by minor amputation. A secondary aim was to identify risk factors for major amputation and death following minor amputation to treat DFD.

### Research design and methods

This was a retrospective analysis of data from patients who required a minor amputation to treat DFD between 2000 and 2019 at a regional tertiary hospital in Queensland, Australia. Baseline characteristics were collected together with remoteness of residence and ethnicity. Remoteness was classified according to the 2019 Modified Monash Model (MMM) system. Ethnicity was based on self-identification as an Aboriginal and Torres Strait Islander or non-Indigenous person. The outcomes of major amputation, repeat minor amputation and death were examined using Cox-proportional hazard analyses.

ethics, governance, and public health act approval by contacting the office of research and innovation at ori@health.qld.gov.au or address GPO Box 48, Brisbane Qld 4001.

**Funding:** This research was supported by grants from the Townsville and Hospital Health Services (SERTA), Tropical Australian Academic Health Centre and Queensland Government. Jonathan Golledge is supported by a Senior Clinical Research Fellowship from the Queensland Government and research grants from the Medical Research Futures Fund and Heart Foundation.

**Competing interests:** The authors have declared that no competing interests exist.

## Results

A total of 534 participants were included, with 306 (57.3%) residing in metropolitan or regional centres, 228 (42.7%) in rural and remote communities and 144 (27.0%) were Aboriginal or Torres Strait Islander people. During a median (inter quartile range) follow-up of 4.0 (2.1–7.6) years, 103 participants (19.3%) had major amputation, 230 (43.1%) had repeat minor amputation and 250 (46.8%) died. The risks (hazard ratio [95% CI]) of major amputation and death were not significantly higher in participants residing in rural and remote areas (0.97, 0.67–1.47; and 0.98, 0.76–1.26) or in Aboriginal or Torres Strait Islander people (HR 1.44, 95% CI 0.96, 2.16 and HR 0.89, 95% CI 0.67, 1.18). Ischemic heart disease (IHD), peripheral artery disease (PAD), osteomyelitis and foot ulceration (p<0.001 in all instances) were independent risk factors for major amputation.

## Conclusion

Major amputation and death are common following minor amputation to treat DFD and people with IHD, PAD and osteomyelitis have an increased risk of major amputation. Aboriginal and Torres Strait Islander People and residents of remote areas were not at excess risk of major amputation.

## Introduction

Diabetes-related foot disease (DFD), such as foot ulceration, infection and gangrene, affects approximately one-third of patients with diabetes over their lifetime [1, 2]. Minor amputation (distal to the ankle joint) is frequently required in the treatment of DFD [3, 4]. Individuals that undergo minor amputations are reported to have reduced quality of life [5, 6] and frequently require further amputation [7].

Multiple studies in countries around the world have reported that re-amputation is common in patients with DFD. A recent meta-analysis estimated that the rate of re-amputation in patients with DFD was 19% (inter quartile range, IQR, 5% to 32%) at one year and 37% (IQR, 27 to 47%) at 5 years following a minor amputation [8]. Furthermore, this previous review found a lack of studies with longer follow-up [8]. A US study reported a major amputation rate of 10% within one year of an ipsilateral toe amputation [9] and a recent meta-analysis estimated that 30% (95% CI 24% to 37%) of people undergoing trans-metatarsal amputation later required a major amputation [10]. Patients requiring a minor amputation also have a risk of mortality of approximately 10% per year [11, 12]. A previous study reported that risk factors for a repeat amputation include chronic obstructive pulmonary disease, peripheral artery disease (PAD), elevated white cell count and previous revascularisation [13], but further studies in other populations were needed.

A recent population cohort study in Italy reported a 4-year mortality of 65% in patients having a minor amputation. Risk factors for mortality were age ≥65 years, cardiovascular disease, and chronic renal disease [14]. A recent study from Australia reported that long distance from the nearest high risk foot clinic and low toe pressures were risk factors for lower extremity amputation [15]. Most studies have found that the burden of DFD is substantially greater in rural and remote and First Nation's populations [16, 17]. Aboriginal and Torres Strait Islander Australians have been reported to have a 3-6-fold increased likelihood of developing a DFD complication compared to non-Indigenous Australians [18].

No prior study has investigated whether Aboriginal and Torres Strait Islander People or residents of more remote regions have an increased risk of major amputation following a minor amputation to treat DFD. The primary aim of this study was to examine the associations between ethnicity and remoteness of residency with the risk of major amputation and death following initial treatment of DFD by minor amputation. A secondary aim was to identify risk factors for major amputation and death following minor amputation to treat DFD.

## Materials and methods

### Study design and data source

This was a retrospective analysis of prospectively collected data of patients who were admitted between the 1st of January 2000 and 31st of December 2019 to a regional tertiary hospital in Queensland Australia (Townsville University Hospital, TUH) for a minor amputation to treat DFD. Patients' records were accessed to obtain information on possible events that occurred between 1st of January 2000 and 31st of December 2020. Data collection was conducted between 1st of December 2022 and 31st of December 2022. Ethics approval was obtained from the Townsville Hospital and Health Service Human Research Ethics Committee [HREC/13/QTHS/125] [19, 20]. This included approval for a patient consent waiver that was required because of the retrospective design. A five-member Aboriginal and Torres Strait Islander reference committee was consulted for the approval of the research as previously described [21]. All data were crossed checked by an experienced data base manager against the patients' records.

### Population

All patients were identified through Operating Rooms Management System (ORMIS); a system that records all surgical procedures conducted in TUH. For inclusion, participants had to have undergone a minor amputation (i.e. an amputation distal to the ankle joint [3]) as treatment for DFD during the study period. Participants also had to be previously diagnosed with diabetes and have data available on place of residence and Aboriginal and Torres Strait Islander ethnicity. The first minor amputation that occurred during the study period between 1st of January 2000 to 31st of December 2019 was defined as the index amputation. Patients with previous major amputation (unilateral or bilateral), those who underwent an amputation procedure in the absence of diabetes, those who were <18 years of age and those with missing patient records were excluded. After identification of the participant through ORMIS, the relevant participant's hospital admission record during which the participant had the index minor amputation procedure was accessed to obtain the necessary data. Subsequent records were also obtained to identify further outcomes.

Minor amputation was defined as amputation distal to the ankle joint [3]. Toe amputations were defined as amputation of a toe at or distal to the metatarso-phalangeal joint. A trans-metatarsal amputation was defined as an amputation across metatarsal bones. Forefoot amputation was defined as an amputation distal to the ankle joint at or proximal to tarso-metatarsal joints [3]. Participants were identified using ORMIS as previously described [16]. All patients undergoing any amputation procedure in the TUH are recorded in this system.

### Risk factors

Participants' baseline risk factors were obtained from hospital admission records at the time of the index minor amputation. These included age, sex, smoking, diabetes (type 1 or type 2), hypertension, ischaemic heart disease (IHD), PAD, end stage renal failure (ESRF), Aboriginal and Torres Strait Islander ethnicity and past history of minor amputation. Current smoking

was defined as active smoking within the last month as documented at the time of the index admission [19]. Diabetes and hypertension were defined by a documented medical diagnosis at the time of hospital admission [19]. Duration of diabetes, insulin treatment and glycaemic level at the time of surgery were not collected as these data were not consistently available in all patients. IHD was defined as a documented history of myocardial infarction, angina or previous treatment of IHD [22]. PAD was defined as ankle brachial pressure index (ABPI) <0.9, previous peripheral revascularisation and/ or imaging identified stenosis or occlusion of lower limb arteries, as previously described [22]. ESRF was defined as requirement for dialysis. The primary reason for the index hospital admission was classified as foot ulcer, soft tissue infection, osteomyelitis or gangrene. In instances where there was a combination of these presentations all relevant data were included. Aboriginal and Torres Strait Islander ethnicity was based on self-identification by patients at the time of the index minor amputation.

## Remoteness

Remoteness was classified according to the 2019 Modified Monash Model (MMM) classification [23] using the post code of the participants, which has previously been associated with the degree of inequality of access to healthcare [24]. There were no participants from MMM1 category. Regional areas, such as Townsville and Mackay, were classified as MMM2, and other regional areas, medium-sized towns, small towns, remote and very remote towns were classified as MMM categories 3 to 7 respectively. For analyses MMM categories 1 to 2 and 3 to 7 were separately grouped because of unequal and skewed distribution of patients across remoteness categories.

## Assessment of outcomes

The primary outcome was major amputation, defined as amputation proximal to the ankle, in either leg during the follow-up period [25]. Other outcomes included requirement for repeat minor amputation in either leg and, all-cause mortality. Outcome events were obtained through ORMIS and review of participants' medical records. The electronic medical records included admissions to all Queensland public hospitals. Participants were censored at the time of their death, or the date of last follow-up identified by the last in- or out-patient hospital record if no event was experienced.

## Sample size calculation

Since this study was a retrospective observational study, the sample size was informed by the number of participants available rather than a pre-specified sample size estimate. Prior to analysis the available sample size was considered based on the plan to assess the association of remoteness with the requirement for major amputation in an adjusted Cox proportional hazard analysis. The Cox proportional hazard analysis was planned to include up to 10 covariates of which some of the variates such as MMM Classification had multiple permutations. Based on a prior study it was estimated that the major amputation rate would be at least 40% during a minimum two-year median follow-up [7]. Based on this, at least 250 individuals would provide a well powered analysis considering the requirement to attain at least 10 outcome events per degree of freedom according to Monte Carlo simulations [26].

## Statistical analyses

Histograms, skewness and kurtosis tests suggested that continuous data were not normally distributed. Thus, continuous data were presented as median and inter-quartile range and

compared between groups using the Mann-Whitney U test. Nominal and categorical data were presented as counts and percentage and compared using the Pearson's chi-squared test. Multivariate Cox proportional hazard analyses were performed to assess the association between MMM categories and major amputation, repeat minor amputations and all cause death after adjusting for age, sex, smoking, IHD, PAD, ESRF and osteomyelitis. Similar analyses were conducted separately for Aboriginal and Torres Strait Islander ethnicity and major amputation, repeat minor amputations and all cause death after adjusting for age, sex, smoking, IHD, PAD, ESRF and osteomyelitis. Selection of variables for adjustment was based on those that have been established as risk factors for events and variables which bivariate comparisons between groups suggested significant differences ($p<0.05$). All model assumptions were met. Factors that were highly correlated were not included in the same model. Cox regression model outcomes were reported as hazard ratios (HR) and 95% confidence intervals (CI). The ability of the model to predict the respective outcomes was assessed using concordance index (c-index). Individual contributions of predictor variables were assessed by calculating the Likelihood Ratio Test (LRT) statistic. LRT was calculated as twice the difference of log likelihood values of the full cox regression model versus a reduced model where one predictor variable was removed to determine its contribution to the outcome prediction. The significance was determined by comparing the obtained LRT values against the difference in degrees of freedom in the critical value of Chi-squared distribution. In the analyses, since one variable was removed at a time, the change in degrees of freedom was 1 corresponding to critical value of 3.84 in the chi-squared distribution table. LRT values greater than the critical value were considered to be significantly important in predicting the outcomes. Kaplan-Meier curves were performed to graphically represent the probability of events over time and statistically represented using log rank tests to compare the events between the different population groups. Data were analysed using SPSS v29 (IBM, Armonk, NY) software package.

## Results and discussion

### Characteristics of participants

A total of 534 participants were included following an index admission for 504 toe amputations (94.4%), 24 trans-metatarsal amputations (4.5%) and 6 mid-tarsal amputations (1.1%). Three hundred and six participants (57.3%) resided in MMM categories 1 or 2, 228 (42.7%) in MMM categories 3–7. One hundred and forty-four (27.0%) participants identified as Aboriginal and Torres Strait Islander Peoples.

Aboriginal and Torres Strait Islanders people were significantly more likely to be residing in MMM categories 3–7 than non-Indigenous participants (Table 1).

Other baseline demographic factors or risk factors were not different between those presenting from MMM categories 1, 2 and those presenting from MMM categories 3–7. Aboriginal and Torres Strait Islander participants were also significantly younger, more likely to be male, have ESRF and be admitted with gangrene, ulceration or infection, but less likely to have had previous revascularisation surgery compared to non-Indigenous participants (Table 2).

Median (IQR) follow-up for the cohort was 4.0 (2.1 to 7.6) years. One hundred and three participants (19.3%) had a major amputation and 230 (43.1%) had a repeat minor amputation. A total of 110 major amputations and 340 minor amputations were performed. Eighty-four participants (81.6%) underwent a major amputation on the ipsilateral side while nineteen (18.4%) had a major amputation on the contralateral side. A total of 250 participants (46.8%) died during follow up. There were no significant differences in the rates of major amputation, minor amputation or mortality between participants from regional cities (MMM categories 1 to 2) and those from more rural localities (MMM categories 3–7), (log rank test p >0.05 in all

**Table 1. Risk factors at recruitment in participants who were admitted for an index minor amputation.**

| Risk factor | All patients (n = 534) | MMMC 1&2 Urban or Regional centres | MMMC 3–7 Regional, rural and remote towns | Significance |
|---|---|---|---|---|
| | | (n = 306) | (n = 228) | |
| Age | 61.61 [53.0–71.0] | 61.00 [52.00–71.00] | 62.00 [53.00–72.00] | 0.406 |
| Male sex | 370 (69.3%) | 215 (70.3%) | 155 (68.0%) | 0.572 |
| Aboriginal and Torres Strait Islander People | 144 (27.0%) | 70 (22.9%) | 74 (32.5%) | **0.014**** |
| Current smoking | 282 (52.8%) | 161 (52.6%) | 121 (53.1%) | 0.917 |
| Hypertension | 391 (73.2%) | 233 (76.1%) | 158 (69.3%) | 0.077 |
| IHD | 217 (40.6%) | 124 (40.5%) | 93 (40.8%) | 0.951 |
| PAD | 190 (35.6%) | 102 (33.3%) | 88 (38.6%) | 0.209 |
| ESRF | 54 (10.1%) | 31 (10.1%) | 23 (10.1%) | 0.987 |
| Past history of minor amputation | 41 (7.7%) | 23 (7.5%) | 18 (7.9%) | 0.871 |
| Past history of revascularisation | | | | |
| Endovascular | 70 (13.1%) | 38 (12.4%) | 32 (14.0%) | 0.584 |
| Open vascular | 70 (13.1%) | 34 (11.1%) | 36 (15.8%) | 0.113 |
| Immediate presenting problem | | | | |
| Osteomyelitis | 146 (27.3%) | 90 (29.4%) | 56 (24.6%) | 0.214 |
| Gangrene | 106 (19.9%) | 61 (19.9%) | 45 (19.7%) | 0.955 |
| Ulcer | 434 (81.3%) | 253 (82.7%) | 181 (81.6%) | 0.335 |
| Infection | 446 (83.5%) | 260 (85.0%) | 186 (81.6%) | 0.297 |

Foot note

Modified Monash Model category, MMMC; ischaemic heart disease, IHD; peripheral artery disease, PAD; end stage renal failure, ESRF.

instances) (Fig 1). Outcomes were similar for Aboriginal and Torres Strait Islander and non-Indigenous Australians (log rank test p values >0.05 in all instances) (Fig 2).

## Association of remoteness and major amputation

Multiple cox regression models were developed (Table 1). Unadjusted analyses suggested no association between remoteness of residence and risk of major amputation (HR 0.97, 95% CI 0.67 to 1.47, p = 0.966). After adjusting for age, sex, smoking, IHD, PAD, ESRF, osteomyelitis and foot ulceration these results remained unchanged (HR 0.96, 95% CI 0.64 to 1.42, p = 0.826) (S1 Table). LRT of the risk factors included in the model suggested that IHD, PAD, osteomyelitis and ulceration were significant predictors of major amputation in people living in remote areas (S2 Table). However, the cox regression model did not achieve a good discrimination to predict the outcomes (c-index 0.505, 95% CI 0.442, 0.569) (S3 Table). Visual representation of the model is provided in the Kaplan-Meier curve (Fig 1A) and the analysis suggested that there was no significant difference in major amputation rates between participants resident in more and less remote locations (Log rank test p value = 0.966).

## Association of Aboriginal and Torres Strait Islander ethnicity and major amputation

Unadjusted analyses suggested no significant association between Aboriginal and Torres Strait Islander ethnicity and risk of major amputation (HR 1.44, 95% CI 0.96 to 2.16, p = 0.078). After adjusting for age, sex, smoking, IHD, PAD, ESRF, osteomyelitis and foot ulceration the results were unchanged (HR 1.29, 95% CI 0.83–2.00, p = 0.096) (S4 Table). LRT of the risk factors

**Table 2. Characteristics of Aboriginal and Torres Strait Islander and non-Indigenous participants.**

|  | Aboriginal and Torres Strait Islander People | Non-Indigenous patients | Significance |
|---|---|---|---|
|  | (n = 144) | (n = 390) |  |
| Age | 54.17 [52.23–56.11] | 64.40 [63.13–65.64] | **<0.001** |
| Male sex | 80 (55.6%) | 290 (74.4%) | **<0.001** |
| Rurality |  |  | **<0.001** |
| MMMC 1 and 2 | 70 (48.6%) | 236 (60.5%) |  |
| MMMC 3 to7 | 74 (51.4%) | 154 (39.5%) |  |
| Current smoking | 80 (55.6%) | 202 (51.2%) | 0.440 |
| Hypertension | 109 (75.7%) | 282 (72.3%) | 0.433 |
| IHD | 51 (35.4%) | 166 (42.6%) | 0.136 |
| PAD | 48 (33.3%) | 142 (36.4%) | 0.510 |
| ESRF | 24 (16.7%) | 30 (7.7%) | **0.002** |
| Past history of minor amputation | 13 (9.0%) | 28 (7.2%) | 0.477 |
| Past history of revascularisation |  |  |  |
| Endovascular | 10 (6.9%) | 60 (15.4%) | **0.010** |
| Open vascular | 9 (6.3%) | 61 (15.6%) | **0.004** |
| Immediate presenting problem |  |  |  |
| Osteomyelitis | 47 (32.6%) | 99 (25.4%) | 0.095 |
| Gangrene | 19 (13.2%) | 87 (22.3%) | **0.019** |
| Ulcer | 126 (87.5%) | 308 (79.0%) | **0.025** |
| Infection | 144 (89.6%) | 317 (81.3%) | **0.022** |

Foot note

Modified Monash Model category, MMMC; ischaemic heart disease, IHD; peripheral artery disease, PAD; end stage renal failure, ESRF.

included in the model suggested that IHD, PAD, osteomyelitis and foot ulceration were significant predictors of major amputation in Aboriginal and Torres Strait Islander people (S5 Table). However, the cox regression model did not achieve a good discrimination to predict the outcomes (c-index 0.541, 95% CI 0.477, 0.606). Visual representation of the model is provided in Fig 2A and the analysis suggested that there was no significant difference in major amputation rates in Aboriginal and Torres Strait Islander and non-Indigenous people (Log rank test p value = 0.076).

## Risk factors for repeat minor amputation

None of the risk factors were independently significantly associated with the risk of repeat minor amputation (S6 Table).

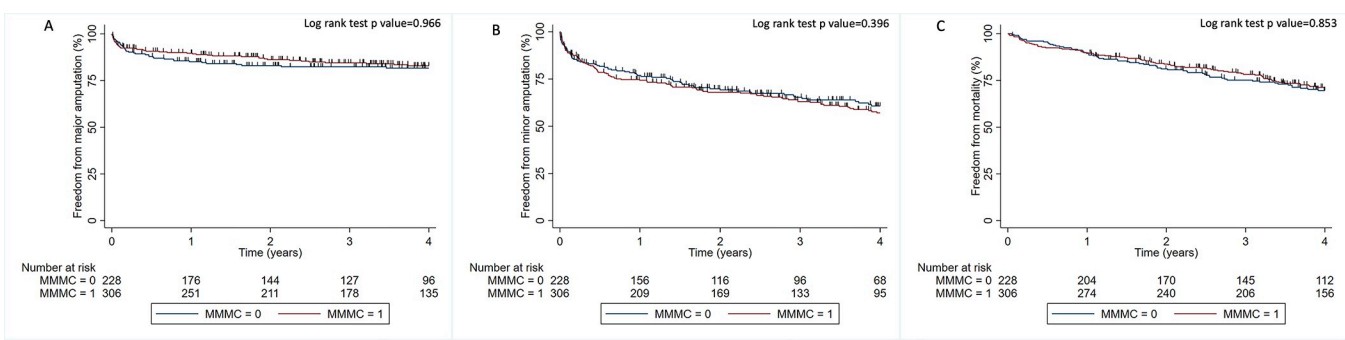

**Fig 1. Kaplan-Meier curve illustrating the freedom from events in participants presenting from large towns or cities (MMMC = 1) and small towns or regional towns (MMMC = 0).** A; Major amputation, B; Minor amputation, C; All-cause mortality. MMMC: Modified Monash Model Classification.

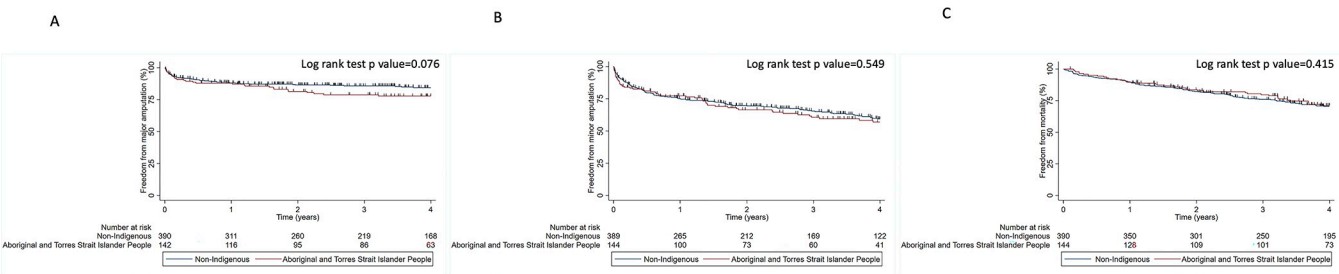

**Fig 2. Kaplan-Meier curve illustrating the freedom from events in Aboriginal and Torres Strait Islander and non-Indigenous participants.** A; Major amputation, B; Minor amputation, C; All-cause mortality.

### Risk factors for deaths

None of the risk factors were independently significantly associated with the risk of death (S6 Table).

## Discussion

This study is the first to assess the impact of remoteness of residence on the outcome of people having minor amputation to treat DFD. Similar studies have been conducted in patients with other forms of vascular disease such as abdominal aortic aneurysms and peripheral artery disease but not in people with DFD [24]. The setting in North Queensland, Australia was well suited for this investigation because the population is widely dispersed with patients presenting from both regional and rural areas with considerable travel time to TUH [24]. This study included 534 participants with 27% identifying as Aboriginal and/or Torres Strait Islander People. One fifth of the participants underwent a major amputation and about half of participants died during a median follow-up of 4 years. Outcomes were not significantly different for Aboriginal and Torres Strait Islander and non-Indigenous people, which may reflect equivalent care given to participants presenting from diverse backgrounds [27]. This is in contrast to previous studies that showed poorer outcomes among Aboriginal and Torres Strait Islanders compared to non-Indigenous patients [18, 28]. IHD, PAD, presence of osteomyelitis and foot ulceration during the index minor amputation were independent predictors of major amputation, as previously reported in other studies [16, 17].

More than one million people worldwide undergo a lower extremity amputation each year to treat DFD [29]. Such amputations cause significant morbidity and mortality [8, 30]. A recent systematic review reported that global rates of hospital admission for all DFD conditions are considerably greater than those for amputations alone [31]. In the USA the rate of non-traumatic limb amputation increased by 50% between 2009 and 2015 [32]. A repeat amputation rate of 26% within 1-year of initial amputation was reported in a study of 71300 participants [30]. Despite the availability of studies investigating the overall rates of amputation [33], there have been no studies conducted to look at subsequent major amputation following an index minor amputation in Australia.

In the current study, during a median follow-up period of 4 years 47% of the participants died and 43% had a repeat minor amputation emphasising the high morbidity and mortality associated with DFD. DFD causes substantial social, psychological, and physical burden [34, 35]. Similar results have been reported in US populations [36]. This highlights the need for better secondary prevention, which is challenging to implement in patients presenting from vast geographic areas, especially in rural populations where access to health services is limited. In the current study participants with IHD, PAD, osteomyelitis and foot ulceration had an increased risk of major

amputation, as has been previously reported from multiple studies [37, 38]. Patients with PAD were twice as likely to have a major amputation as compared to those with no PAD, similar to findings from previous studies [16, 21]. A previous study from North Queensland found an excess of distal tibial artery disease which can be challenging to revascularise [39]. The distal nature of the artery disease and the challenges of implementing medical management may have contributed to the high amputation rates reported [39]. This study found that ESRF was also an independent risk factor for major amputation, as previously reported [40]. The association of IHD with major amputation may be secondary to undetected PAD in these participants [41].

A number of previous studies have reported an excess of major amputations in rural locations [42]. In contrast, this study found that there is no significant difference in the rates of amputation in participants living in remote locations compared to those living in larger towns. This finding maybe reflective of state-wide expansion in podiatry care provides to rural locations, along with use of telehealth services for endocrinology introduced in Queensland [28].

The current study found that the rates of major amputation were not significantly different in non-Indigenous and Australian First Nations patients. The unadjusted HR did however suggest a non-significant (p = 0.078) 1.44-fold excess risk of major amputation in Aboriginal and Torres Strait Islander compared to non-Indigenous patients. The lack of significant difference in outcomes may be reflect multiple health promotion programs that were implemented by the government to close the healthcare gap or may simply reflect insufficient sample size to adequately compare outcomes [42]. The findings of this study highlight the need for continued efforts to reduce the morbidity and mortality related to DFD, via frequent podiatry care and improved medical management. Use of telehealth services and community-based podiatry is recommended to improve access to services in rural localities [43]. Early referral of patients to tertiary care centres is recommended for early interventions to prevent major amputation by revascularisation procedures.

The current study has a number of strengths and limitations. Strengths included the relatively large sample size and the inclusion of substantial numbers of Aboriginal and Torres Strait Islander participants. Limitations include the retrospective study design and the lack of information on the Wound Ischaemia Foot Infection (WIFI) classification and use of revascularisation procedures. It is also possible that some outcome evets may have been missed if participants moved outside the North Queensland region. It is also important to note that these findings may not be generalisable to other regions where delivery of health care and composition of the population may be different.

## Conclusions

Major amputation and death are very common following minor amputation to treat DFD. Remoteness was not associated with an increased risk of major amputation which may be attributable to equity of healthcare delivered to different populations. Risk of major amputation was increased in participants with IHD, PAD, osteomyelitis and foot ulceration.

## Supporting information

**S1 Table. Association of remoteness with major amputation following a minor amputation to treat diabetes-related foot disease.**
(DOCX)

**S2 Table. Contribution of individual risk factors for the overall cox regression model that included remoteness.**
(DOCX)

**S3 Table. Contribution of individual risk factors for the overall cox regression model that included Aboriginal and Torres Strait Islander ethnicity.**
(DOCX)

**S4 Table. Association of ethnicity with major amputation following a minor amputation to treat diabetes-related foot disease.**
(DOCX)

**S5 Table. C statistic or the area under the curve of receiver operating characteristic curves for each risk factor and its ability to predict the outcome of a major amputation among participants who underwent a minor amputation following diabetes-related foot disease.**
(DOCX)

**S6 Table. Cox proportional hazard analyses for the association between Modified Monash Model (MMC) categories and Aboriginal and Torres Strait Islander status versus repeat minor amputation and death.**
(DOCX)

**S1 File. Inclusivity in global research.**
(DOCX)

## Acknowledgments

The authors would like to acknowledge all the staff of Queensland Research Centre for Peripheral Vascular Disease.

## Author Contributions

**Conceptualization:** Chanika Alahakoon, Shivshankar Thanigaimani, Aaron Drovandi, Malindu Fernando, Peter A. Lazzarini, Joseph V. Moxon, Jonathan Golledge.

**Data curation:** Chanika Alahakoon.

**Formal analysis:** Chanika Alahakoon, Shivshankar Thanigaimani, Tejas P. Singh, Joseph V. Moxon.

**Funding acquisition:** Jonathan Golledge.

**Investigation:** Chanika Alahakoon.

**Methodology:** Chanika Alahakoon, Shivshankar Thanigaimani, Tejas P. Singh, Aaron Drovandi, Malindu Fernando, Peter A. Lazzarini, Jonathan Golledge.

**Project administration:** Jonathan Golledge.

**Resources:** Chanika Alahakoon, James Charles, Peter A. Lazzarini, Jonathan Golledge.

**Software:** Chanika Alahakoon, Tejas P. Singh.

**Supervision:** Shivshankar Thanigaimani, Malindu Fernando, Peter A. Lazzarini, Joseph V. Moxon, Jonathan Golledge.

**Validation:** Chanika Alahakoon, James Charles, Jonathan Golledge.

**Visualization:** Chanika Alahakoon, Tejas P. Singh.

**Writing – original draft:** Chanika Alahakoon.

**Writing – review & editing:** Chanika Alahakoon, Shivshankar Thanigaimani, Tejas P. Singh, Aaron Drovandi, James Charles, Malindu Fernando, Peter A. Lazzarini, Joseph V. Moxon, Jonathan Golledge.

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
