## [Decision Letter · Decision Letter 0]

11 Dec 2023

PONE-D-23-28544Association between remoteness and ethnicity with major amputation following minor amputation in people with diabetes-related foot diseasePLOS ONE

Dear Dr. Golledge,

Thank you for submitting your manuscript to PLOS ONE. After careful consideration, we feel that it has merit but does not fully meet PLOS ONE’s publication criteria as it currently stands. Therefore, we invite you to submit a revised version of the manuscript that addresses the points raised during the review process.

We look forward to receiving your revised manuscript.

Kind regards,

Moin Uddin Ahmed

Academic Editor

PLOS ONE

 “This research was supported by grants from the Townsville and Hospital Health Services (SERTA), Tropical Australian Academic Health Centre and Queensland Government. Jonathan Golledge is supported by a Senior Clinical Research Fellowship from the Queensland Government and research grants from the Medical Research Futures Fund and Heart Foundation.”

Additional Editor Comments:

Thank you for submitting your manuscript to PLOS One. The peer review process has provided valuable insights through the feedback from our three diligent reviewers. We appreciate your dedication to advancing the quality of your work. I kindly request that you provide a comprehensive response to all the comments provided by the three reviewers. Please address each point raised with clarity, indicating the specific changes made or providing a rationale if any suggestions were not implemented. This thorough response will greatly facilitate the editorial evaluation and expedite the final decision.

We look forward to receiving your revised manuscript and accompanying response at your earliest convenience. Should you have any questions or require further clarification, do not hesitate to contact us.

Reviewers' comments:

Reviewer's Responses to Questions

**Comments to the Author**

1. Is the manuscript technically sound, and do the data support the conclusions?

Reviewer #1: Partly

Reviewer #2: Yes

Reviewer #3: Yes

2. Has the statistical analysis been performed appropriately and rigorously? 

Reviewer #1: I Don't Know

Reviewer #2: Yes

Reviewer #3: Yes

3. Have the authors made all data underlying the findings in their manuscript fully available?

Reviewer #1: Yes

Reviewer #2: Yes

Reviewer #3: Yes

4. Is the manuscript presented in an intelligible fashion and written in standard English?

Reviewer #1: Yes

Reviewer #2: Yes

Reviewer #3: Yes

5. Review Comments to the Author

Reviewer #1: Comments to the authors:

Overall impression

This could have been an important piece of work highlighting inequities between Indigenous vs non Indigenous Australians, but several key opportunities to do relevant analyses and/or structure arguments that would contribute to future policy or practice implications are missed.

Major comments

1. The “key message” section needs reworking – the questions asked are not adequately answered

2. P2, Line 37 – What is already known? “single centre retrospective cohort study” does not answer the question

3. P2, Line 29 – What this study adds? The authors have provided a description of study results and does not answer the question

4. P2, Line 45 – how might this study affect research practice and policy making? Again, a summary of results are provided, not the application the results to practice and policy.

Abstract

1. Introduction – please provide some background on the issue

2. P4, Line 76, two “and”s

Main text

Introduction

General comment: the intro needs reworking. The bulk of the intro is a description of numerous studies and little attempt at synthesis of the literature (i.e consistencies and inconsistencies, gaps etc). Aboriginal and Torres Strait Islanders have poorer outcomes in DFD in Australia compared to non Indigenous people – is there any literature of major after minor amputations in non Indigenous Australians? What is the comparator?

There can be a greater emphasis on poor outcomes in Indigenous vs non Indigenous Australians and I think the authors have missed an opportunity to highlight this in the intro

Additional: P5, Line 92 – formatting needed for interquartile range, IQR

Research design and methods

1. P6, Line 128 – patients were followed up until 31/12/2020 – if this is a retrospective analysis, how did you follow patients up? Did you call patients at the end of the study period to check if they had a major amputation? How are you accounting for differing length in follow up period if you are?

2. P7 Line 133 – A 5 member Aboriginal and Torres Strait Islander reference committee was consulted for approval of the research as previously described (20) – is this a follow up of another study, or is this a stand alone retrospective analysis of data? It is unclear from this statement and methods

3. P7 Line 136 – inconsistent with results – did you collect data only if the amputation was their first ever amputation (referred to as Index amputation in results section) or just if they had one amputation during this time? This is an important distinguishing factor

4. P7 Line 141 – ORMIS abbreviation needs expanding

5. P7 Potential risk factors – should work into introduction the literature behind each individual risk factor. More description also needed about “diabetes” – type of diabetes? Duration? Insulin dependence? Glycemia at time of surgery?

6. Remoteness – I don’t understand the rationale behind dichotomizing remoteness 1 and 2 vs 3 to 7. The authors potentially lose granularity and important differences between the more remote areas which could provide interesting analyses and important policy implications. If there was a rationale behind the dichotomization (i.e unequal or skewed distribution of participants across remoteness categories) this is important for readers to understand

7. The authors extracted data on past history of revascularization (endovascular vs open) – are there more details on duration, current toe pressures/ABI etc which may be more relevant?

8. Did authors attempt to extract data on reason for future amputation?

Results

1. Table 1 – should present whole of cohort/population demographics before dichotomizing between MMMC classification

2. P13, Line 247 – median follow up was 4 (2.1 – 7.6 years). Is this just from the time of amputation to end of study? Were you able to check all records that all participants were still residing in catchment area and not relocated elsewhere to another health service?

3. Did you look at death/mortality as a competing risk for major amputation? Considering half died in the follow up period

4. 50% mortality is a major finding and consistent with other international publications on post-op mortality following minor amputation – a missed opportunity to elaborate on this and analyse difference in mortality between Indigenous vs non Indigenous people

Discussion

The discussion is haphazard and misses the opportunity to provide coherent arguments towards poorer outcomes in Indigenous Australians/Australians residing remotely. Instead it repeats the results and adds findings from a few other studies (which would be better placed in the Intro)

PAD is a known risk factor for major amputation and this study does not add anything to this knowledge

Line 336 – the authors did not state earlier the reasons for poor revascularization in Indigenous people – is this an access issue? Cultural? The assumption that this is due to distal/tibial vessel disease is not scientifically sound

Reviewer #2: Thank you for the invitation to review this paper. Overall this is a well written piece of work and will make a useful addition to the literature in Australia on diabetes related foot disease.

This was a retrospective study conducted over a ten year period of n= 534 participants aiming to examine the association between remoteness with risk of 1 major amputation, repeat minor amputation and death following initial treatment of DFD by a minor amputation in North Queensland, Australia. There are also a number of secondary outcomes including exploring the relationship between Aboriginality and the same outcomes.

The key messages should be reviewed and care taken in answering these with greater accuracy as the current responses are not accurate and do not reflect the key messages of this work, in my opinion.

Abstract

Introduction could use a little more ‘so what’ in the introduction. Why is this important? This is a multidisciplinary journal so it needs to be put into context for the readers rather then simply outlining the aims of the work.

The introduction

There is a nice global overview of research presented but a notable lack of research conducted in Australia. Given this is an Australian paper in an Australian population, I would appreciate more Australian research to be included. I found this paper below which is related on minor amputation that could be considered for inclusion, which looks at engagement with podiatry services in people who undergo minor ampuattion.

Linton, C., Searle, A., Hawke, F., Tehan, P. E., & Chuter, V. (2021). Nature and extent of outpatient podiatry service utilisation in people with diabetes undergoing minor foot amputations: a retrospective clinical audit. Journal of Foot and Ankle Research, 14(1), 1-6.

Similarly, other Australian researchers have conducted work which would be valuable to include either in the background or the discussion – this one in particular I thought was relevant looking at discharges from hospital from different local areas in the context of socioeconomic status, which is also considered a predictor of amputation?

Bergin, S. M., Brand, C. A., Colman, P. G., & Campbell, D. A. (2011). The impact of socio-economic disadvantage on rates of hospital separations for diabetes-related foot disease in Victoria, Australia. Journal of foot and ankle research, 4, 1-6.

Methods

Some further clarification may be useful here to allow for replication.

How was data extracted from the medical record? Was it using ICD codes or reporting or was it manually extracted? I am assuming maybe you pulled ICD codes for minor amputation then extracted data based on medical record numbers. How did you ensure there was a diabetes diagnosis? Was the information extracted by one person? Was there a process for checking data for accuracy?

Results

Your results, particularly in relation to Aboriginal and Torres Strait Islander people are an alarming reflection on the gap in outcomes that are still being experienced in this population in Australia. In your one-way analysis, you demonstrate that Aboriginal people with DFD are significantly younger, more likely to be female compared to non-aboriginal people, significantly more ESRF – also significantly less likely to have had surgical intervention for PAD despite similar rates of PAD. Is this worthy of further discussion? I appreciate it would not be through a deficit lens as such, but rather a call to action on there is more important work to be done to close this gap? This may not be appropriate so would appreciate the expertise of your group to determine.

Discussion

A somewhat similar study was conducted in another part of Australia which would be useful to bring in to the discussion – as some of your findings were similar, and some different. I believe this would really enrich the discussion. Whilst this work was not specifically following minor amputation, it did look at predictors of amputation in an Australian cohort with foot disease, and there are some similarities in findings, namely PAD and infection were associated with increased odds of amputation, however they also found that distance from specialised care WAS associated with increased risk of amputation also. This difference may be nice to tease out in the discussion.

Tehan, P. E., Hawes, M. B., Hurst, J., Sebastian, M., Peterson, B. J., & Chuter, V. H. (2022). Factors influencing lower extremity amputation outcomes in people with active foot ulceration in regional Australia: A retrospective cohort study. Wound Repair and Regeneration, 30(1), 24-33.

Overall this is an important piece of work that has been well executed and is worthy of publication. I wish the authors every success in their important future work.

Reviewer #3: Thank you for the opportunity to review this paper. The paper presents important data related to longer term outcomes post minor amputation, which is important now that minor amputation is considered a form of 'treatment' for DFD and is performed in such great numbers. The data related to indigenous status and remoteness as a risk factor for further amputation is also important and is a gap in our current knowledge. The study is very well constructed and presents reliable and valuable data. The paper is extremely well written and I have no recommendations for changes.

6. PLOS authors have the option to publish the peer review history of their article (what does this mean?). If published, this will include your full peer review and any attached files.

Reviewer #1: No

Reviewer #2: No

Reviewer #3: **Yes: **Shan Bergin

---

## [Author Response · Author response to Decision Letter 0]

17 Jan 2024

Reviewer #1: Comments to the authors:

Overall impression

This could have been an important piece of work highlighting inequities between Indigenous vs non-Indigenous Australians, but several key opportunities to do relevant analyses and/or structure arguments that would contribute to future policy or practice implications are missed.

Thank you for taking your time to review our manuscript. We have made changes to accommodate your comments and improve it.

Major comments

1. The “key message” section needs reworking – the questions asked are not adequately answered

2. P2, Line 37 – What is already known? “single centre retrospective cohort study” does not answer the question

3. P2, Line 29 – What this study adds? The authors have provided a description of study results and does not answer the question

4. P2, Line 45 – how might this study affect research practice and policy making? Again, a summary of results are provided, not the application the results to practice and policy.

Thank you for the above comments. We have changed the whole “Key Message” section to read as follows:

KEY MESSAGE

What is already known?

Toe or minor amputation is common in patients with diabetes-related foot disease (DFD).

What this study adds?

Five hundred and 34 patients who had a minor amputation for DFD were follow-up for a median of 4 years during which time 19% had a major amputation and 47% died. There was no significant association between remoteness of residency or ethnicity and the risk of major amputation or death. 

How might this study affect research practice and policy making?

This study suggested that there is a substantial risk of major amputation and death in people who have had a minor amputation to treat DFD, highlighting the need for better secondary prevention. Aboriginal and Torres Strait Islander Peoples and residents in rural and remote locations did not have a significantly higher risk of major amputation and death compared to non-Indigenous people living in less remote locations, suggesting previously identified unequal outcomes may have been addressed.

Abstract

1. Introduction – please provide some background on the issue

Thank you for the comment. We have modified the abstract as follows:

ABSTRACT

Introduction

Minor amputation is commonly needed to treat diabetes-related foot disease (DFD). The primary aim of this study was to examine the associations between ethnicity and remoteness of residency with the risk of major amputation and death following initial treatment of DFD by minor amputation. A secondary aim was to identify risk factors for major amputation and death following minor amputation to treat DFD.

Research design and methods

This was a retrospective analysis of data from patients who required a minor amputation to treat DFD between 2000 and 2019 at a regional tertiary hospital in Queensland, Australia. Baseline characteristics were collected together with remoteness of residence and ethnicity. Remoteness was classified according to the 2019 Modified Monash Model (MMM) system. Ethnicity was based on self-identification as an Aboriginal and Torres Strait Islander or non-Indigenous person. The outcomes of major amputation, repeat minor amputation and death were examined using Cox-proportional hazard analyses.

Results

A total of 534 participants were included, with 306 (57.3%) residing in metropolitan or regional centres, 228 (42.7%) in rural and remote communities and 144 (27.0%) were Aboriginal or Torres Strait Islander people. During a median (inter quartile range) follow-up of 4.0 (2.1-7.6) years, 103 participants (19.3%) had major amputation, 230 (43.1%) had repeat minor amputation and 250 (46.8%) died. The risks (hazard ratio [95% CI]) of major amputation and death were not significantly higher in participants residing in rural and remote areas (0.97, 0.67-1.47; and 0.98, 0.76-1.26) or in Aboriginal or Torres Strait Islander people (HR 1.44, 95% CI 0.96, 2.16 and HR 0.89, 95% CI 0.67, 1.18). Ischemic heart disease (IHD), peripheral artery disease (PAD), osteomyelitis and foot ulceration (p<0.001 in all instances) were independent risk factors for major amputation.

Conclusion

Major amputation and death are common following minor amputation to treat DFD and people with IHD, PAD and osteomyelitis have an increased risk of major amputation. Aboriginal and Torres Strait Islander People and residents of remote areas were not at excess risk of major amputation.

2. P4, Line 76, two “and”s

We made this change. The abstract is given above, and the change was made to the last sentence of the Results section of the Abstract.

Results

A total of 534 participants were included, with 306 (57.3%) residing in metropolitan or regional centres, 228 (42.7%) in rural and remote communities and 144 (27.0%) were Aboriginal or Torres Strait Islander people. During a median (inter quartile range) follow-up of 4.0 (2.1-7.6) years, 103 participants (19.3%) had major amputation, 230 (43.1%) had a repeat minor amputation and 250 (46.8%) died. The risk (hazard ratio [95% CI]) of major amputation and death were not significantly higher in participants residing in rural and remote areas (0.97, 0.67-1.47; and 0.98, 0.76-1.26) or in Aboriginal or Torres Strait Islander people (HR 1.44, 95% CI 0.96, 2.16 and HR 0.89, 95% CI 0.67, 1.18). Ischemic heart disease (IHD), peripheral artery disease (PAD), osteomyelitis and foot ulceration (p<0.001 in all instances) were independent risk factors for major amputation.

Main text

Introduction

General comment: the intro needs reworking. The bulk of the intro is a description of numerous studies and little attempt at synthesis of the literature (i.e consistencies and inconsistencies, gaps etc). Aboriginal and Torres Strait Islanders have poorer outcomes in DFD in Australia compared to non Indigenous people – is there any literature of major after minor amputations in non Indigenous Australians? What is the comparator? 

There can be a greater emphasis on poor outcomes in Indigenous vs non Indigenous 

Australians and I think the authors have missed an opportunity to highlight this in the intro

Additional: P5, Line 92 – formatting needed for interquartile range, IQR

We were unable to find similar studies looking at major amputations following an index minor amputation in the Australian setting. But we included some studies from the Australian setting and changed the Introduction as you suggested. 

We changed it as suggested. The Introduction now reads as follows:

INTRODUCTION

Diabetes-related foot disease (DFD), such as foot ulcers, infections, and gangrene, affects approximately one-third of patients with diabetes over their lifetime (1, 2). Minor amputation (distal to the ankle joint) is frequently required in the treatment of DFD (3, 4). Individuals that undergo minor amputations are reported to have reduced quality of life (5, 6) and frequently require further amputation (7). 

Multiple studies in countries around the world have reported that re-amputation is common in patients with DFD. A recent meta-analysis estimated that the rate of re-amputation in patients with DFD was 19% (inter quartile range, IQR, 5% to 32%) at one year and 37% (IQR, 27 to 47%) at 5 years following a minor amputation (8). Furthermore, this previous review found a lack of studies with longer follow-up (8). A US study reported a major amputation rate of 10% within one year of an ipsilateral toe amputation (9) and a recent meta-analysis estimated that 30% (95% CI 24% to 37%) of people undergoing trans-metatarsal amputation later required a major amputation (10). Patients requiring a minor amputation also have a risk of mortality of approximately 10% per year (11, 12). A previous study reported that risk factors for a repeat amputation include chronic obstructive pulmonary disease, peripheral artery disease (PAD), elevated white cell count and previous revascularisation (13), but further studies in other populations were needed.

A recent population cohort study in Italy reported a 4-year mortality of 65% in patients having a minor amputation. Risk factors for mortality were age ≥65 years, cardiovascular disease, and chronic renal disease (14). A recent study from Australia reported that long distance from the nearest high risk foot clinic and low toe pressures were risk factors for lower extremity amputation (15). Most studies have found that the burden of DFD is substantially greater in rural and remote and First Nation’s populations (16, 17). Aboriginal and Torres Strait Islander Australians have been reported to have a 3-6-fold increased likelihood of developing a DFD complication compared to non-Indigenous Australians (18). 

No prior study has investigated whether Aboriginal and Torres Strait Islander People or residents of more remote regions have an increased risk of major amputation following a minor amputation to treat DFD. The primary aim of this study was to examine the associations between ethnicity and remoteness of residency with the risk of major amputation and death following initial treatment of DFD by minor amputation. A secondary aim was to identify risk factors for major amputation and death following minor amputation in people with DFD. 

Research design and methods

1. P6, Line 128 – patients were followed up until 31/12/2020 – if this is a retrospective analysis, how did you follow patients up? Did you call patients at the end of the study period to check if they had a major amputation? How are you accounting for differing length in follow up period if you are?

We accessed all patients’ records until 31/12/2020 through the hospital data base which is common to the region. Therefore, even if the patient was admitted to another hospital, the event will be captured. Different length of follow up was addressed through conducting a cox regression analysis instead of a logistic regression analysis which will not take the follow up period into account. 

We changed the methods section to clarify this issue and it now reads as follows:

RESEARCH DESIGN AND METHODS

Study design and data source

This was a retrospective analysis of prospectively collected data of patients who were admitted between the 1st of January 2000 and 31st of December 2019 to a regional tertiary hospital in Queensland Australia (Townsville University Hospital, TUH) for a minor amputation to treat DFD. Patients’ records were accessed to obtain information on possible events that occurred between 1st of January 2000 to 31st of December 2020. Data collection was conducted between 1st of December 2022 to 31st of December 2022. Ethics approval was obtained from the Townsville Hospital and Health Service Human Research Ethics Committee [HREC/13/QTHS/125](19, 20). This included approval for a patient consent waiver that was required because of the retrospective design. A five-member Aboriginal and Torres Strait Islander reference committee was consulted for the approval of the research as previously described (21). All data were crossed checked by an experienced database manager against the patients’ records.

2. P7 Line 133 – A 5 member Aboriginal and Torres Strait Islander reference committee was consulted for approval of the research as previously described (20) – is this a follow up of another study, or is this a stand alone retrospective analysis of data? It is unclear from this statement and methods

This is not a follow up of another study. This is a stand-alone study conducted on patients with diabetes and subsequent minor amputations. There may be an overlap of these patents with some other studies from our group as the approval was taken in the initial stages under a broader umbrella to look at outcomes in patients with any vascular surgical problems (PAD, DFD) between Aboriginal and Torres Strait Islander people vs non-Indigenous Australians.

3. P7 Line 136 – inconsistent with results – did you collect data only if the amputation was their first ever amputation (referred to as Index amputation in results section) or just if they had one amputation during this time? This is an important distinguishing factor

The first minor amputation recorded between 2000 to 2019 was taken as the index amputation. It is not the first ever amputation as some patients have undergone a previous minor amputation which occurred before the 1st of January 2000. 

We have clarified this, and the Methods section now reads as follows:

Population

For inclusion, participants had to have undergone a minor amputation to treat DFD during the study period, been previously diagnosed with diabetes and have data available on place of residence and Aboriginal and Torres Strait Islander ethnicity. The first minor amputation that occurred during the study period was defined as the index amputation. Patients with previous major amputation (unilateral or bilateral), those who underwent an amputation procedure in the absence of diabetes, those who were <18 years of age and those with missing patient records were excluded. After identification of the participant through Operating Rooms Management System (ORMIS), the relevant participant’s hospital admission record during which the participant had the index minor amputation was accessed to obtain the necessary data. 

Minor amputation was defined as amputation distal to the ankle joint (3). Toe amputations were defined as amputation of a toe at or distal to the metatarso-phalangeal joint. A trans-metatarsal amputation was defined as an amputation across metatarsal bones. Forefoot amputation was defined as an amputation distal to the ankle joint at or proximal to tarso-metatarsal joints (3). Participants were identified using ORMIS as previously described (16). All patients undergoing any amputation procedure in the TUH are recorded in this system. 

4. P7 Line 141 – ORMIS abbreviation needs expanding

ORMIS stands for: Operating Rooms Management System. We have corrected this mistake and it appears as follows:

For inclusion, participants had to have undergone a minor amputation to treat DFD during the study period, been previously diagnosed with diabetes and have data available on place of residence and Aboriginal and Torres Strait Islander ethnicity. The first minor amputation that occurred during the study period was defined as the index amputation. Patients with previous major amputation (unilateral or bilateral), those who underwent an amputation procedure in the absence of diabetes, those who were <18 years of age and those with missing patient records were excluded. After identification of the participant through Operating Rooms Management System (ORMIS), the relevant participant’s hospital admission record during which the participant had the index minor amputation was accessed to obtain the necessary data. 

5. P7 Potential risk factors – should work into introduction the literature behind each individual risk factor. More description also needed about “diabetes” – type of diabetes? Duration? Insulin dependence? Glycemia at time of surgery?

We included both type 1 and type 2 diabetic patients. Median duration of time was not available in most patients. Insulin dependency and glycaemia at the time of surgery was also not consistent withing patient charts. We included this in the methods. 

Risk factors 

Participants’ baseline risk factors were obtained from hospital admission records at the time of the index minor amputation. These included age, sex, smoking, diabetes (type 1 or type 2), hypertension, ischaemic heart disease (IHD), PAD, end stage renal failure (ESRF), Aboriginal and Torres Strait Islander ethnicity and past history of minor amputation. Current smoking was defined as active smoking within the last month as documented at the time of the index admission (22). Diabetes and hypertension were defined by a documented medical diagnosis at the time of hospital admission (22). Duration of diabetes, insulin treatment and glycaemic level at the time of surgery were not collected as these data were not consistently available in all patients. IHD was defined as a documented history of myocardial infarction, angina or previous treatment of IHD (23). PAD was defined as ankle brachial pressure index (ABPI) <0.9, previous peripheral revascularisation and/ or imaging identified stenosi

---

## [Decision Letter · Decision Letter 1]

19 Mar 2024

PONE-D-23-28544R1Association of remoteness and ethnicity with major amputation following minor amputation to treat diabetes-related foot disease.PLOS ONE

Dear Dr. Golledge,

Thank you for submitting your manuscript to PLOS ONE. After careful consideration, we feel that it has merit but does not fully meet PLOS ONE’s publication criteria as it currently stands. Therefore, we invite you to submit a revised version of the manuscript that addresses the points raised during the review process.

**Please kindly address some of the comment although both reviewers have recommended acceptance. **

The research methodology is fundamentally solid, but a more detailed exposition of the recruitment process would enhance the paper's clarity and robustness. Certain sections, particularly the Abstract, exhibit a lack of cohesion and fail to articulate a compelling argument for the study's significance. Moreover, the discussion of the findings, especially considering their divergence from previous research on Australian Indigenous populations with Diabetic Foot Disease (DFD), lacks depth. The paper does not sufficiently explore the implications of these findings for future funding, service access for remote populations, and the overall management of DFD in Indigenous communities.

The presentation of the findings appears somewhat cursory, with little consideration of their broader impact or how they might inform future clinical practices for DFD across all populations. This oversight suggests a missed opportunity to leverage the collected data for substantive clinical improvements.

We look forward to receiving your revised manuscript.

Kind regards,

Yee Gary Ang, MBBS MPH

Academic Editor

PLOS ONE

Journal Requirements:

Reviewers' comments:

Reviewer's Responses to Questions

**Comments to the Author**

1. If the authors have adequately addressed your comments raised in a previous round of review and you feel that this manuscript is now acceptable for publication, you may indicate that here to bypass the “Comments to the Author” section, enter your conflict of interest statement in the “Confidential to Editor” section, and submit your "Accept" recommendation.

Reviewer #2: All comments have been addressed

Reviewer #3: (No Response)

2. Is the manuscript technically sound, and do the data support the conclusions?

Reviewer #2: Yes

Reviewer #3: Yes

3. Has the statistical analysis been performed appropriately and rigorously? 

Reviewer #2: Yes

Reviewer #3: Yes

4. Have the authors made all data underlying the findings in their manuscript fully available?

Reviewer #2: Yes

Reviewer #3: Yes

5. Is the manuscript presented in an intelligible fashion and written in standard English?

Reviewer #2: Yes

Reviewer #3: Yes

6. Review Comments to the Author

Reviewer #2: (No Response)

Reviewer #3: (No Response)

7. PLOS authors have the option to publish the peer review history of their article (what does this mean?). If published, this will include your full peer review and any attached files.

Reviewer #2: No

Reviewer #3: No

---

## [Author Response · Author response to Decision Letter 1]

27 Mar 2024

27-03-2024

Dr Yee Gary Ang

Academic Editor

PLOS ONE

Title of the manuscript: 

PONE-D-23-28544R1

Association of remoteness and ethnicity with major amputation following minor amputation to treat diabetes-related foot disease.

PLOS ONE

Thank you for accepting this manuscript for publication. We wish to resubmit this manuscript after making the changes the reviewers suggested. The detail comments are shown below.

We hope to hear from you soon.

Disclosure: The funders had no role in study design, data collection and analysis, decision to publish, or preparation of the manuscript.

Thank you.

Professor Jonathan Golledge.

Please kindly address some of the comment although both reviewers have recommended acceptance. 

1) The research methodology is fundamentally solid, but a more detailed exposition of the recruitment process would enhance the paper's clarity and robustness. 

Thank you for the comment.

The “Population” section of our Materials and Methods has now been changed to read as follows.

Population

All patients were identified through Operating Rooms Management System (ORMIS); a system that records all surgical procedures conducted in TUH. For inclusion, participants had to have undergone a minor amputation (i.e. an amputation distal to the ankle joint [3]) as treatment for DFD during the study period. Participants also had to be previously diagnosed with diabetes and have data available on place of residence and Aboriginal and Torres Strait Islander ethnicity. The first minor amputation that occurred during the study period between 1st of January 2000 to 31st of December 2019 was defined as the index amputation. Patients with previous major amputation (unilateral or bilateral), those who underwent an amputation procedure in the absence of diabetes, those who were <18 years of age and those with missing patient records were excluded. After identification of the participant through ORMIS, the relevant participant’s hospital admission record during which the participant had the index minor amputation procedure was accessed to obtain the necessary data. Subsequent records were also obtained to identify further outcomes. 

Minor amputation was defined as amputation distal to the ankle joint [3]. Toe amputations were defined as amputation of a toe at or distal to the metatarso-phalangeal joint. A trans-metatarsal amputation was defined as an amputation across metatarsal bones. Forefoot amputation was defined as an amputation distal to the ankle joint at or proximal to tarso-metatarsal joints [3]. Participants were identified using ORMIS as previously described [16]. All patients undergoing any amputation procedure in the TUH are recorded in this system. 

2) Certain sections, particularly the Abstract, exhibit a lack of cohesion and fail to articulate a compelling argument for the study's significance. 

Thank you for the Comment. The abstract has been changed to read as follows.

Abstract

Introduction

Minor amputation is commonly needed to treat diabetes-related foot disease (DFD). Remoteness of residence is known to limit access to healthcare and has previously been associated with poor outcomes. The primary aim of this study was to examine the associations between ethnicity and remoteness of residency with the risk of major amputation and death following initial treatment of DFD by minor amputation. A secondary aim was to identify risk factors for major amputation and death following minor amputation to treat DFD.

Research design and methods.

This was a retrospective analysis of data from patients who required a minor amputation to treat DFD between 2000 and 2019 at a regional tertiary hospital in Queensland, Australia. Baseline characteristics were collected together with remoteness of residence and ethnicity. Remoteness was classified according to the 2019 Modified Monash Model (MMM) system. Ethnicity was based on self-identification as an Aboriginal and Torres Strait Islander or non-Indigenous person. The outcomes of major amputation, repeat minor amputation and death were examined using Cox-proportional hazard analyses.

Results

A total of 534 participants were included, with 306 (57.3%) residing in metropolitan or regional centres, 228 (42.7%) in rural and remote communities and 144 (27.0%) were Aboriginal or Torres Strait Islander people. During a median (inter quartile range) follow-up of 4.0 (2.1-7.6) years, 103 participants (19.3%) had major amputation, 230 (43.1%) had repeat minor amputation and 250 (46.8%) died. The risks (hazard ratio [95% CI]) of major amputation and death were not significantly higher in participants residing in rural and remote areas (0.97, 0.67-1.47; and 0.98, 0.76-1.26) or in Aboriginal or Torres Strait Islander people (HR 1.44, 95% CI 0.96, 2.16 and HR 0.89, 95% CI 0.67, 1.18). Ischemic heart disease (IHD), peripheral artery disease (PAD), osteomyelitis and foot ulceration (p<0.001 in all instances) were independent risk factors for major amputation.

Conclusion

Major amputation and death are common following minor amputation to treat DFD and people with IHD, PAD and osteomyelitis have an increased risk of major amputation. Aboriginal and Torres Strait Islander People and residents of remote areas were not at excess risk of major amputation.

3) Moreover, the discussion of the findings, especially considering their divergence from previous research on Australian Indigenous populations with Diabetic Foot Disease (DFD), lacks depth. The paper does not sufficiently explore the implications of these findings for future funding, service access for remote populations, and the overall management of DFD in Indigenous communities.

The presentation of the findings appears somewhat cursory, with little consideration of their broader impact or how they might inform future clinical practices for DFD across all populations. This oversight suggests a missed opportunity to leverage the collected data for substantive clinical improvements.

Thank you for the comment. The revised discussion now reads as follows.

Discussion

This study is the first to assess the impact of remoteness of residence on the outcome of people having minor amputation to treat DFD. Similar studies have been conducted in patients with other forms of vascular disease such as abdominal aortic aneurysms and peripheral artery disease but not in people with DFD [25]. The setting in North Queensland, Australia was well suited for this investigation because the population is widely dispersed with patients presenting from both regional and rural areas with considerable travel time to TUH [25]. This study included 534 participants with 27% identifying as Aboriginal and/or Torres Strait Islander People. One fifth of the participants underwent a major amputation and about half of participants died during a median follow-up of 4 years. Outcomes were not significantly different for Aboriginal and Torres Strait Islander and non-Indigenous people, which may reflect equivalent care given to participants presenting from diverse backgrounds [28]. This is in contrast to previous studies that showed poorer outcomes among Aboriginal and Torres Strait Islanders compared to non-Indigenous patients [18, 29]. IHD, PAD, presence of osteomyelitis and foot ulceration during the index minor amputation were independent predictors of major amputation, as previously reported in other studies [16, 17]. 

More than one million people worldwide undergo a lower extremity amputation each year to treat DFD [30]. Such amputations cause significant morbidity and mortality [8, 31]. A recent systematic review reported that global rates of hospital admission for all DFD conditions are considerably greater than those for amputations alone [32]. In the USA the rate of non-traumatic limb amputation increased by 50% between 2009 and 2015 [33]. A repeat amputation rate of 26% within 1-year of initial amputation was reported in a study of 71300 participants [31]. Despite the availability of studies investigating the overall rates of amputation [34], there have been no studies conducted to look at subsequent major amputation following an index minor amputation in Australia. 

In the current study, during a median follow-up period of 4 years 47% of the participants died and 43% had a repeat minor amputation emphasising the high morbidity and mortality associated with DFD. DFD causes substantial social, psychological, and physical burden [35, 36]. Similar results have been reported in US populations [37]. This highlights the need for better secondary prevention, which is challenging to implement in patients presenting from vast geographic areas, especially in rural populations where access to health services is limited. In the current study participants with IHD, PAD, osteomyelitis and foot ulceration had an increased risk of major amputation, as has been previously reported from multiple studies [38, 39]. Patients with PAD were twice as likely to have a major amputation as compared to those with no PAD, similar to findings from previous studies [16, 21]. A previous study from North Queensland found an excess of distal tibial artery disease which can be challenging to revascularise [40]. The distal nature of the artery disease and the challenges of implementing medical management may have contributed to the high amputation rates reported [40]. This study found that ESRF was also an independent risk factor for major amputation, as previously reported [41]. The association of IHD with major amputation may be secondary to undetected PAD in these participants [42].

A number of previous studies have reported an excess of major amputations in rural locations [43]. In contrast, this study found that there is no significant difference in the rates of amputation in participants living in remote locations compared to those living in larger towns. This finding maybe reflective of state-wide expansion in podiatry care provides to rural locations, along with use of telehealth services for endocrinology introduced in Queensland [28]. 

The current study found that the rates of major amputation were not significantly different in non-Indigenous and Australian First Nations patients. The unadjusted HR did however suggest a non-significant (p=0.078) 1.44-fold excess risk of major amputation in Aboriginal and Torres Strait Islander compared to non-Indigenous patients. The lack of significant difference in outcomes may be reflect multiple health promotion programs that were implemented by the government to close the healthcare gap or may simply reflect insufficient sample size to adequately compare outcomes [43]. The findings of this study highlight the need for continued efforts to reduce the morbidity and mortality related to DFD, via frequent podiatry care and improved medical management. Use of telehealth services and community-based podiatry is recommended to improve access to services in rural localities [44]. Early referral of patients to tertiary care centres is recommended for early interventions to prevent major amputation by revascularisation procedures. 

The current study has a number of strengths and limitations. Strengths included the relatively large sample size and the inclusion of substantial numbers of Aboriginal and Torres Strait Islander participants. Limitations include the retrospective study design and the lack of information on the Wound Ischaemia Foot Infection (WIFI) classification and use of revascularisation procedures. It is also possible that some outcome evets may have been missed if participants moved outside the North Queensland region. It is also important to note that these findings may not be generalisable to other regions where delivery of health care and composition of the population may be different.

---

## [Editor Report · Decision Letter 2]

28 Mar 2024

Association of remoteness and ethnicity with major amputation following minor amputation to treat diabetes-related foot disease.

PONE-D-23-28544R2

Dear Dr. Golledge,

We’re pleased to inform you that your manuscript has been judged scientifically suitable for publication and will be formally accepted for publication once it meets all outstanding technical requirements.

Kind regards,

Yee Gary Ang, MBBS MPH

Academic Editor

PLOS ONE

---

## [Editor Report · Acceptance letter]

2 Jun 2024

PONE-D-23-28544R2 

PLOS ONE

Dear Dr. Golledge, 

I'm pleased to inform you that your manuscript has been deemed suitable for publication in PLOS ONE. Congratulations! Your manuscript is now being handed over to our production team.

Kind regards, 

on behalf of

Dr. Yee Gary Ang 

Academic Editor

PLOS ONE